# Lichen-Derived Diffractaic Acid Inhibited Dengue Virus Replication in a Cell-Based System

**DOI:** 10.3390/molecules28030974

**Published:** 2023-01-18

**Authors:** Naphat Loeanurit, Truong Lam Tuong, Van-Kieu Nguyen, Vipanee Vibulakhaophan, Kowit Hengphasatporn, Yasuteru Shigeta, Si Xian Ho, Justin Jang Hann Chu, Thanyada Rungrotmongkol, Warinthorn Chavasiri, Siwaporn Boonyasuppayakorn

**Affiliations:** 1Center of Excellence in Applied Medical Virology, Department of Microbiology, Faculty of Medicine, Chulalongkorn University, Bangkok 10330, Thailand; 2Interdisciplinary Program in Microbiology, Graduate School, Chulalongkorn University, Bangkok 10330, Thailand; 3Center of Excellence in Natural Products Chemistry, Department of Chemistry, Faculty of Science, Chulalongkorn University, Bangkok 10330, Thailand; 4Department of Medicinal Chemistry, Faculty of Pharmacy, University of Medicine and Pharmacy at Ho Chi Minh City, Ho Chi Minh City 700000, Vietnam; 5Institute of Fundamental and Applied Sciences, Duy Tan University, Ho Chi Minh City 710000, Vietnam; 6Faculty of Natural Sciences, Duy Tan University, Da Nang 550000, Vietnam; 7Department of Biology, Faculty of Science, Chulalongkorn University, Bangkok 10330, Thailand; 8Center for Computational Sciences, University of Tsukuba, 1-1-1 Tennodai, Tsukuba 305-8577, Ibaraki, Japan; 9Laboratory of Molecular RNA Virology and Antiviral Strategies, Department of Microbiology and Immunology, Yong Loo Lin School of Medicine, National University of Singapore, Singapore 117545, Singapore; 10Program in Bioinformatics and Computational Biology, Graduate School, Chulalongkorn University, Bangkok 10330, Thailand; 11Biocatalyst and Environmental Biotechnology Research Unit, Department of Biochemistry, Faculty of Science, Chulalongkorn University, Bangkok 10330, Thailand

**Keywords:** dengue virus, depside, depsidone, diffractaic acid, drug discovery

## Abstract

Dengue is a mosquito-borne flavivirus that causes 21,000 deaths annually. Depsides and depsidones of lichens have previously been reported to be antimicrobials. In this study, our objective was to identify lichen-derived depsides and depsidones as dengue virus inhibitors. The 18 depsides and depsidones of *Usnea baileyi*, *Usnea aciculifera*, *Parmotrema dilatatum*, and *Parmotrema tsavoense* were tested against dengue virus serotype 2. Two depsides and one depsidone inhibited dengue virus serotype 2 without any apparent cytotoxicity. Diffractaic acid, barbatic acid, and Parmosidone C were three active compounds further characterized for their efficacies (EC_50_), cytotoxicities (CC_50_), and selectivity index (SI; CC_50_/EC_50_). Their EC_50_ (SI) values were 2.43 ± 0.19 (20.59), 0.91 ± 0.15 (13.33), and 17.42 ± 3.21 (8.95) μM, respectively. Diffractaic acid showed the highest selectivity index, and similar efficacies were also found in dengue serotypes 1–4, Zika, and chikungunya viruses. Cell-based studies revealed that the target was mainly in the late stage with replication and the formation of infectious particles. This report highlights that a lichen-derived diffractaic acid could become a mosquito-borne antiviral lead as its selectivity indices ranged from 8.07 to 20.59 with a proposed target at viral replication.

## 1. Introduction

Dengue infection has become a global burden that causes an estimated 100 million cases and 21,000 deaths annually, according to the World Health Organization (WHO) 2022. Clinical manifestations include high fever, myalgia, thrombocytopenia, plasma leakage, hemorrhage, hypovolemic shock, and multiple organ failure. The causative agent is dengue virus consisting of four serotypes (DENV1–4). Evidence suggests that DENV2 is likely correlated to severity [1,2]. DENV1–4 are members of the family *Flaviviridae* consisting of a 10 kb single-stranded RNA genome enclosed in an ER-derived envelope. Transmission is caused by a mosquito bite, and the virus infects local dermal cells, including dendritic cells or antigen-presenting cells. Infected dendritic cells travel to the lymphatic and blood circulation and then infect remote organs, causing systemic infection. Secondary heterotypic infection or secondary infection with a different DENV serotype contributes to pathologically robust but incompetent immunity with excessive proinflammatory cytokines. The primary treatment is still supportive, and no specific drug is currently available. The only commercial vaccine is dengvaxia, which the CDC recommended for 9–16 years old with a laboratory-confirmed previous DENV infection and living in an endemic area. Drug candidates such as ivermectin and celgosivir failed the proof-of-concept antiviral efficacies in clinical trials as their effective dosage exceeded the pharmacokinetic dosage [3].

Lichens are known for their medicinal properties across cultures, especially in temperate and arctic climates [4,5]. The fungi and algae/cyanobacteria symbiont produce chemically distinct secondary metabolites [6]. However, few scientific reports have been found regarding their scarcity in nature, sophisticated extraction techniques, and low yield [7,8,9,10]. Depsides and depsidones are secondary metabolites found in lichens. Their core structures are composed of orcinol [7]. Depsides and depsidones with various biological activities have been reported, including antioxidant [8,9], antimicrobial [10,11], antiparasitic [12], anticancer [13,14], and antiviral [15,16,17]. A previous study showed that atranorin and its derivatives exhibited anti-hepatitis C virus (HCV) activity in the entry and replication stage [17]. Since DENV and HCV belong to the family *Flaviviridae*, these compounds could potentially inhibit DENV. In this study, we screened 18 depsides and depsidones extracted from lichens and reported that diffractaic acid was the most potent inhibitor of DENV, in addition to inhibiting other mosquito-borne viruses.

## 2. Results

### 2.1. Preliminary Screening, Efficacies, and Cytotoxicities in DENV-2-Infected Cells

Two depsides and 14 depsidones were initially tested against DENV2 New Guinea C (NGC) strain in Vero cells with a compound concentration of 10 µM. Two depsides, diffractaic acid (**1a**) and barbatic acid (**1b**), showed DENV2 inhibition at 99.98% ± 0.04% and 99.99% ± 0.02%, respectively, without apparent cytotoxicity (Figure 1, Table 1). Furthermore, three depsidones, parmosidone C (**2a**), galbinic acid (**3a**), and norstictic acid (**3b**), were less potent at 96.48% ± 2.90%, 80.50% ± 4.85%, and 78.00% ± 7.55%, respectively. All compounds did not show apparent toxicity under the tested conditions (10 µM compound in 1% DMSO), suggesting that depsides and depsidones were not toxic to Vero cells (Table 1). The three compounds (**1a**, **1b**, and **2a**) with at least 90% inhibition were then further analyzed for efficacies (EC_50_) and cytotoxicities (CC_50_). The results showed that diffractaic acid (**1a**) had the highest selectivity index of 20.59 with an EC_50_ of 2.43 ± 0.19 µM (Table 1, Appendix A) and CC_50_ of 50.13 ± 7.45 µM (Table 1, Appendix A). The higher selectivity index indicated the potentially broader therapeutic range in subsequent studies. Barbatic acid (**1b**) was more effective than diffractaic acid with a lower EC_50_ at 0.91 ± 0.15 µM (Table 1, Appendix A). However, the barbatic acid was highly cytotoxic with the CC_50_ of 12.10 ± 0.38 µM (Table 1, Appendix A) and a selectivity index of 13.33. Furthermore, parmosidone C (**2a**) was the least effective of the three compounds with the EC_50_ at 17.42 ± 3.21 µM (Table 1, Appendix A) and the least toxic with the CC_50_ at 155.83 ± 7.77 µM (Table 1, Appendix A). Furthermore, the efficacy of diffractaic acid was confirmed in a human hepatoma cell line (Huh-7) infected with DENV2 16681. The results showed a similar efficacy at an EC_50_ of 3.89 ± 0.07 µM (Table 1, Appendix A), while Huh-7 cells were less toxic than Vero cells with a CC_50_ >100 µM (Table 1, Appendix A), and the selectivity index was >25. Therefore, diffractaic acid was selected as a representative compound for further cytotoxic studies and efficacies in other mosquito-borne viruses and an enterovirus.

### 2.2. Cytotoxicities and Efficacies of Diffractaic Acid in other Human-Derived Cell Lines and Mosquito-Borne Viruses

Next, cytotoxicity was studied in other human-derived cell lines, including HepG2, HEK-293, THP-1, and RD cells (Table 1, Appendix A). Half-maximal cytotoxic concentration (CC_50_) varied between approximately 40 and 70 µM with the lowest and highest CC_50_ in HepG2 and HEK-293 cells, respectively (Table 1, Appendix A). HepG2, HEK-293, and THP-1 cells represented hepatocytes, renal epithelia, and monocytes that were generally susceptible to dengue and other mosquito-borne viruses and were naturally targets after the primary viremia. RD cells originated from striated muscles that were susceptible to enterovirus A71. The results showed that diffractaic acid displayed mild to moderate cytotoxicity in the mammalian cells.

The efficacies against four serotypes of dengue viruses (DENV1, 2, 3, and 4), Zika virus (ZIKV), and chikungunya virus (CHIKV) were carried out in Vero cells (Table 1, Appendix A), while that of enterovirus A71 (EV-A71) was studied in RD cells (Appendix A). The compound inhibited DENV1-4 and ZIKV with values of EC_50_ similar to DENV2 (EC_50_ range: 2.43–4.90 µM) (Table 1, Appendix A). However, the compound was less effective against CHIKV (EC_50_ of 6.21 ± 0.69 µM) (Table 1, Appendix A) and poorly inhibited EV-A71 (EC_50_ of 19.93 ± 1.94 µM) (Table 1, Appendix A). Therefore, diffractaic acid effectively inhibited mosquito-borne flaviviruses (DENV1-4 and ZIKV) and also inhibited a mosquito-borne alphavirus (CHIKV).

### 2.3. Diffractaic Acid Inhibits DENV2 after Entry and Interferes with Viral Replication

The molecular target of diffractaic acid was screened using a time-of-addition assay with low viral titer to initially determine whether the target resides at early or late timepoints. Diffractaic acid at 10 µM was added to DENV2-infected Vero cells (MOI of 0.1) at various timepoints (0, 2, 4, 6, 8, 10, 12, 24, 36, 48, 60, and 72 h post infection) (Figure 2A). DMSO was used as a control without inhibition. After 72 h, the supernatants were collected, and viral inhibition was determined by plaque titration. The result showed that viral titer significantly decreased by 4 log, 3 log, and 2 log in the periods of 2–10, 12–60, and 72 h post infection (hpi), respectively (Figure 2B). In addition, little effect was observed at 0 h after infection (Figure 2B). At this timepoint, the compound was added and removed along with DENV2 infection, suggesting that the compound did not inhibit viral attachment and entry. An additional attachment inhibition was performed, and the results confirmed that post-infection inhibition was the most effective mode (Figure 2C,D). In this experiment, the temperature was brought to 4 °C during 1 h infection (Figure 2C). The results showed that pre-incubation and co-incubation did not contribute to any significant viral inhibition (Figure 2D). Furthermore, the compound did not inhibit pH-dependent fusion mediated by conformational change of the envelope proteins (Appendix A). It is possible that the molecular target was host-derived. Moreover, the compound effectively inhibited late timepoints (12–60 h, Figure 2B), suggesting that later steps in the viral life cycle could be affected, and the compound was stable throughout the experiment.

To further determine the dynamic effect of diffractaic acid on a single round of DENV2 infection, time-of-addition (TOA) and time-of-removal (TOR) assays were established (Figure 3A,B). Briefly, compound at 10 µM was added to or removed from DENV2-infected Vero cells (MOI of 1) at the respective time points (0, 6, 12, 24, 36, and 48 hpi). Supernatants and cell lysates were collected for viral progeny, genome, and protein analysis using the plaque assay, RT-qPCR, and Western blotting, respectively (Figure 3C–E). DMSO was used as a no inhibition control (Appendix A). The results showed that the compound effectively inhibited virion progeny when added as late as 36 h after infection (Figure 3C, black circle). In other words, the compound was immediately effective whenever it was introduced into DENV2-infected cells, even up to 36 h after infection. However, the time of removal suggested that the compound did not leave any remaining inhibitory effect when it was removed earlier than collection (Figure 3C, white circle). Only removal at 48 h or immediately before collection showed inhibition of the virion. In other words, the inhibitory effect was removed whenever the compound was removed from the infected cells. Therefore, the molecular target of diffractaic acid is expected to be abundant and easily renewable in cells.

Furthermore, the viral genome levels of the compound-treated group were reported as a proportion of the corresponding DMSO controls (Figure 3D). The results showed that the viral genome pattern was more sensitive than those of the vial progeny. Genome levels were able to detect the inverse correlation pattern between TOA and TOR at 36 h (Figure 3D). This result suggested that viral replication should be the primary site targeted by the compound. Protein expression appeared minimally altered during TOA and TOR courses (Figure 3E, Appendix A), implying that the viral protein was insensitive to the dynamics of the compound. To conclude, the viral genome was the most sensitive parameter, whereas the viral protein was the least sensitive.

Additionally, the DENV2 16,681 was continuously subpassage under the selective pressure of a sublethal dose of diffractaic acid and DMSO control (Appendix A) before analysis of possible escape mutants using whole-genome sequencing. Unfortunately, a single synonymous nucleotide substitution (G5178A, L219) was discovered. This position belongs to subdomain 1 of NS3 between the ATP- and Mg^2+^-binding sites of the ATPase/RTPase viral enzyme. We conclude that this is likely an accidental finding.

In addition, viral targets related to the viral replication are NS2/3 protease, NS3 helicase, NS5 RdRp, and NS5 MTase. These proteins were set up for molecular docking with diffractaic acid, and the binding energy was compared to the native inhibitor of each target (Figure 4A). Diffractaic acid was bound to the allosteric site of NS2/3 protease, the RNA-binding sites of NS3 helicase and NS5 RdRp, and the SAM-binding site of NS5 MTase. The top pose of 100 runs of molecular docking for each viral target was plotted as shown in Figure 4A. The native inhibitors of NS2/3 protease (compound 9 [18]), NS3 helicase (ST-610 [19]), and NS5 RdRp (NITD-107 [20]) were superior to diffractaic acid. On the contrary, the binding energy of the native inhibitor (sinefungin, −8.9 kcal/mol [21]) overlapped with diffractaic acid in the range of −7.8 to −9.4 kcal/mol. The SAM binding of NS5 MTase is mainly contributed by hydrophobic interaction, involving alkyl–π (K105, H110, F133, I147, R160, and R163), van der Waals (E111, G148, and K180), and anion–π (E149) interactions due to the methyl group and aromatic structure of diffractaic acid (Figure 4B). Additionally, we found two hydrogen bond interactions (D131 and R163). These residues have been reported to contribute a large amount to previously reported inhibitors [22]. Therefore, the methyl moieties of diffractaic acid could play a significant role in antiviral activity by interrupting methyltransferase activity. However, diffractaic acid showed an inhibitory concentration (IC_50_) of 49.30 ± 1.86 µM (Figure 4C) in the methyltransferase study. Therefore, it is unlikely that the flaviviral methyltransferase could potentially be a target. The molecular target remains to be elucidated.

## 3. Discussion

Depsides and depsidones are lichen-derived secondary metabolites with several biological activities such as anticancer, antibacterial, antioxidant, antiparasitic, and antiviral [8,9,10,11,12,13,14,15,16,17]. We extracted, purified, and screened the anti-DENV2 activity of *Usnea baileyi*, *Usnea aciculifera*, and *Parmotrema dilatatum* and found three compounds with inhibitory activity against DENV2 over 90%. Two depsides showed good and similar EC_50_ values, possibly due to similar structures. Parmosidone C was the only depsidone with over 90% antiviral activity, but it was still less effective than the depsides. Moreover, parmosidone C was nontoxic to other mammalian cells (MCF-7, human breast cancer) with IC_50_ over 100 µM [23]. However, according to the EC_50_ result, the depsidone core is unlikely the active candidate for further analysis.

The selectivity index revealed that diffractaic acid was the best among the three compounds (SI = 20.59), being twofold higher than that of barbatic acid. Moreover, a previous report suggested that barbatic acid was cytotoxic to many cancer cells but not to PBMCs [12,13]. Our results showed that barbatic acid was toxic to Vero cells (CC_50_ = 12.10 ± 0.38 µM), corresponding to previous reports [13]. Therefore, diffractaic acid was further investigated for cytotoxicity and broad-spectrum antiviral activity. Diffractaic acid exhibited CC_50_s of 39.32 to >100 µM in HepG2, HEK-293, THP-1, Huh-7m and RD cells. The toxicity was higher in cancer cell lines than normal cell lines (HEK-293, human embryonic kidney cells). Our finding corresponds to previous reports on the cytotoxic activity to cancer cell lines and antitumor activity [24,25,26]. Furthermore, diffractaic acid was used to challenge mice, and the lethal dose (LD_50_) was 962 mg/kg orally [27]. The lethal dose was at least 65 times higher than the CC_50_s. Therefore, diffractaic acid could be less toxic toward animals than observed in the cell-based system.

In addition, diffractaic acid broadly inhibited DENV1–4 and ZIKV with similar EC_50_, was slightly less effective against an enveloped mosquito-borne alphavirus (CHIKV), and was not effective against a non-enveloped enterovirus (EV-A71). However, enterovirus replication is distinct from that seen for DENV, ZIKV, and CHIKV, especially in terms of IRES-dependent translation and the requirement of primers and host-derived factors for replication [28]. Therefore, it is possible that the molecular target could be the viral or host components shared among flaviviruses and an alphavirus.

Time-course experiments and the functional assays revealed that diffractaic acid inhibited DENV virion progeny after entry by interfering with viral replication. The compound could bind reversibly to the target, and viral RNA was the most sensitive parameter. Another lichen-derived depside, atranorin and its derivatives, also potently inhibited the hepatitis C virus, a member of the family *Flaviviridae*, at the entry and replication stage with an efficacy of 11.8–22.3 µM [17]. The compound-driven mutation was a silent mutation that is not located in the ligand-binding site, suggesting that this mutation may be a coincidence. As the compound similarly inhibited both viruses, the distinctive mechanism of flaviviral RNA replications such as protease, helicase, cap formation, and methylation should be primarily considered as potential targets. The pan-docking results suggest that NS5 MTase was the most likely potential target (Figure 4A). The mode of action could involve the methyl moieties of diffractaic acid by interfering with the SAM-binding site of NS5 MTase (Figure 4B), which was a shared location among flaviviruses and the alphavirus. The result is consistent with cell-based efficacies against enteroviruses, since the virus lacked methyltransferase and showed no efficacy. However, the IC_50_ of diffractaic acid against dengue methyltransferase was more than 20 times higher than the EC_50_ in the cell-based assay. Therefore, it is possible that methyltransferase is only a contributing factor. Thus, the major target remains to be explored.

## 4. Materials and Methods

### 4.1. Extraction, Purification, and Identification of Lichen Depsidones and Depsides

The two depsides and 14 depsidones were extracted from *Usnea aciculifera* [29], *Usnea baileyi* [30], *Parmotrema dilatatum* [31], and *Parmotrema tsavoense* [23] (Table 1). The compounds were identified using ^1^H, ^13^C, and 2D NMR methods as described previously. Briefly, air-dried parts of *U. aciculifera* (2.02 kg) were ground using the maceration method at room temperature and extracted with n-hexane, dichloromethane (DCM, CH_2_Cl_2_), ethyl acetate (EtOAc), and methanol (MeOH) (4 × 10 L). The filtered solution was evaporated under reduced pressure to produce the n-hexane extract (50.0 g), DCM extract (101.0 g), EtOAc extract (110.0 g), and MeOH extract (55.0 g), respectively. The DCM extract (101.0 g) was separated by quick column chromatography using gradient elution with n-hexane–EtOAc (stepwise 80:20–0:10) and EtOAc–MeOH (stepwise 10:0–0:10) to give 13 fractions, DCM.1–DCM.13. DCM.1 (28.0 g) was subjected to a silica gel column eluted with n-hexane–EtOAc (85:15) to give seven subfractions 1.1–1.7. Subfraction 1.1 (4.95 g) was chromatographed with a Sephadex LH-20 column (100 g) with MeOH–DCM (60:40), followed by chromatography with normal silica gel eluted with hexane–DCM (60:40) to produce methyl orselinate (**4**, 8.0 mg) and 7-hydroxy-5-methoxy-6-methylphthalide (**5**, 11 mg). Subfraction 1.2 (1.8 g) was chromatographed with the Sephadex LH-20 column (100 g) with MeOH–DCM (60:40) followed by chromatography with RP-C_18_ silica gel eluted with H_2_O–MeOH (60:40) to give norstictic acid (**3b**, 28 mg). Subfraction 1.3 (7.9 g) was subjected to a silica gel column using gradient elution with n-hexane–EtOAc (85:15–0:100) to give barbatic acid (**1b**, 2025.0 mg) and diffractaic acid (**1a**, 2560.0 mg).

The chemical structures of all isolated compounds were elucidated by a combination of spectroscopic data (1D and 2D NMR, HRESIMS), as well as comparison of their NMR data with those of the literature (Table 1 [7,8,9,10]).

All compound stocks were stored in solid powder at room temperature. The compounds were diluted in DMSO (PanReac AppliChem, Hesse, Germany) at a final concentration of 50 mM and stored in an aliquot at −20 °C until use.

### 4.2. Cells and Viruses

Vero (ATCC^®^CCL-81), LLC/MK2 (ATCC^®^CCL-7), C6/36 (ATCC^®^CRL-1660), HepG2 (ATCC^®^HB-8065), HEK-293 (ATCC^®^CRL-1573), and THP-1 (ATCC^®^TIB-202) cells were maintained as previously described [32]. Huh-7 cells (JCRB0403) were maintained as previously described [22]. RD cells (ATCC^®^CRL-1573) were maintained in Dulbecco’s modified Eagle medium (DMEM, Gibco, Langley, VA, USA) supplemented with 10% fetal bovine serum, 100 U/mL penicillin, and 100 µg/mL streptomycin at 37 °C with 5% CO_2_.

Reference strains of DENV1 (16007), DENV2 (New Guinea C; NGC), DENV2 (16681), DENV3 (16562), DENV4 (c0036), ZIKV (SV0127/14), and CHIKV (ECSA genotype) were propagated in C6/36 cells as previously described [32]. EV-A71 (BRCR) was propagated in Vero cells with M199 medium supplemented with 1% fetal bovine serum, 100 IU/mL penicillin, and 100 µg/mL streptomycin at 37 °C with 5% CO_2_.

### 4.3. Efficacy Test

Vero cells (5 × 10^4^) were seeded in a 24-well plate and incubated at 37 °C under 5% CO_2_ overnight. Cells were infected with DENV2 NGC at a multiplicity of infection (MOI) of 0.1, and compounds were added at 10 µM; DMSO at 1% was used as a no inhibition control. Cultures were incubated for 1 h with gentle rocking every 15 min. Cells were washed with PBS, and cultured in MEM supplemented with 1% FBS, 100 IU/mL penicillin, and 100 µg/mL streptomycin; 10 mM HEPES was added in the presence of the compound at 10 µM. Cells were incubated at 37 °C with 5% CO_2_ for 3 days. The supernatants were collected to determine the viral titers using a plaque titration assay as previously described [33]. Compounds that showed an inhibition rate of more than 90% were used for further investigation.

The selected compounds were further analyzed for the effective concentration (EC_50_). Each compound was serially diluted to 10 different concentrations and added to DENV2-infected Vero cells (MOI of 0.1), unless otherwise indicated. Cells were incubated for 3 days, and supernatants were collected for plaque titration assay. The efficacy of each compound was calculated from the fit of the nonlinear regression curve and the concentration required for the reduction of 50% viral titer (EC_50_) was determined. Results were reported as the mean and the standard error of mean (SEM) from three independent experiments.

### 4.4. Time Course Assay

The possible target of the compound was initially screened by a time-of-addition assay [32]. Vero cells (5 × 10^4^) were seeded in a 24-well plate and incubated as described. Cells were infected with DENV2 NGC at an MOI of 0.1 for 1 h with gentle rocking every 15 min. Diffractaic acid at 10 µM was added at various timepoints: 0, 2, 4, 6, 8, 10, 12, 24, 36, 48, 60, and 72 h post infection (hpi). DMSO at 1% was a mock treatment or no inhibition control. After 72 h of incubation, supernatants were collected to determine viral titers using a plaque titration assay. The results were confirmed by three independent experiments.

To further investigate the role of the compound in the interference viral replication cycle, we performed the time-of-addition and time-of-removal assays. Vero cells were infected with DENV2 NGC at an MOI of 1 for 1 h with gentle rocking every 15 min. For the time-of-removal assay, the 10 µM diffractaic acid was added to DENV2-infected cells after infection. The compound was removed at the indicated timepoints after infection by washing with PBS and replacing with maintenance medium. For the time-of-addition assay, 10 µM diffractaic acid was added at the indicated timepoints after infection. DMSO at 1% was used as a mock treatment. After 48 h of incubation, the supernatants were collected to determine viral titers, viral RNA, and viral protein using the plaque titration assay, RT-qPCR, and Western blot, respectively. The results were confirmed by three independent experiments.

### 4.5. Cytotoxicity Test

The cytotoxicity of the compounds was also accessed at 10 µM in parallel with viral inhibition screening. Vero cells (1 × 10^4^) were seeded in a 96-well plate and incubated at 37 °C under 5% CO_2_ overnight. Compounds were added and incubated for 2 days. DMSO at 1% was used as a mock treatment. Cytotoxicity was measured using the CellTiter 96^®^ Aqueous One Solution Cell Proliferation Assay (MTS) kit (Promega, Madison, WI, USA) according to the manufacturer’s instructions and analyzed by spectrophotometry at *A_450 nm_.*

The cytotoxic concentration (CC_50_) of effective compounds was analyzed by diluting to 8–10 concentrations and adding them to the Vero cells, unless otherwise indicated. Cultures were incubated for 2 days. Cytotoxicity was measured using the CellTiter 96^®^ Aqueous One Solution Cell Proliferation Assay (MTS) kit (Promega, Madison, WI, USA) according to the manufacturer’s instructions and analyzed by spectrophotometry at *A_450 nm_.* The cytotoxicity of each compound was calculated from the nonlinear regression curve, and the concentration required for 50% cell death (CC_50_) was determined. Results were reported as the mean and standard error of mean (SEM) from three independent experiments.

### 4.6. Attachment Inhibition Study

Vero cells (5 × 10^4^) were seeded in a 24-well plate and incubated as described. DENV2 NGC (MOI of 0.1) was inoculated into the cells at 4 °C for 1 h. Then, 10 µM diffractaic acid was incubated with DENV2 prior to addition to cells (pre-incubation), co-incubated with DENV2 (co-incubation), or added to DENV2-infected cells after 1 h of incubation (post-attachment incubation). Cultures were maintained for 2 days, and supernatants were collected to determine viral titers by plaque titration assay. The results were confirmed by three independent experiments.

### 4.7. Fusion Inhibition Assay

C6/36 cells (2 × 10^5^) were seeded in a 24-well plate and incubated at 28 °C overnight [34]. Cells were infected with DENV2 NGC (MOI of 1) for 1 h with gentle rocking every 15 min. Then, 10 µM diffractaic acid was added to virus-infected cells during and after infection. The 4G2 antibody and 1% DMSO were used as positive inhibition and no inhibition controls. Cells were incubated for 2 days; 0.5 MES (2-(N-morpholino) ethane sulfonic acid) (Bio Basic Canada, Ontario, Canada) was added to the induced acidic condition (pH 5–6) and then incubated for 1–2 days until fused cells were observed under a light microscope. Cell fusion was observed with a Nikon ECLIPSE TS100 inverted microscope (Nikon, New York, NY, USA).

### 4.8. RNA Extraction and Quantitative RT-PCR (RT-qPCR)

Viral RNA in infected cells was extracted using NucleoZOL™ reagent (MACHEREY NAGEL, Dueren, Germany) and a Direct-zol™ RNA MiniPrep Kit (Zymo Research, Irvine, CA, USA) according to the manufacturer’s instructions. The viral genome in the experiment was determined by quantitative RT-PCR. The RT-qPCR was performed using a Power SYBR^®^ Green RNA-to-CT™ 1-Step kit (Applied Biosystems™, Waltham, MA, USA) with a Step-OnePlus Real-Time PCR System ABI 7500 (Applied Biosystems™, Waltham, MA, USA) as previously described [32]. The β-actin gene was used as an internal control [35].

### 4.9. Western Blotting

The infected cells were lysed using NP-40 lysis buffer (containing 1% NP-40, 50 mM Tris, 150 mM NaCl, 5 mM EDTA, and protease inhibitor cocktails (Himedia, Mumbai, India)) and centrifuged at 12,000× *g* for 20 min to obtain the cell lysate. For DENV2 protein detection, cell lysates were subjected to SDS-PAGE under nonreducing conditions. The 4G2 monoclonal antibody (HB-112, ATCC, Manassas, VA, USA) was used as the primary antibody for the DENV2 E protein. Mouse anti-β-actin antibody (Biolegend, San Diago, CA, USA) was used as a loading control. HRP-conjugated goat anti-mouse IgG antibody was used as a secondary antibody (Biolegend, San Diago, CA, USA). The chemiluminescence signals were developed using Immobilon Classico Western HRP substrate (Merck, Darmstadt, Germany). Protein band intensities were quantified using ImageJ software version 1.53.

### 4.10. Molecular Docking Study

The crystal structures of DENV viral proteins (NS2/3 protease (2FOM [36]), NS3 helicase (2bmf [37]), NS5 RdRp (3VWS [20]), and NS5 MTase (5EHI [38])) were used as a protein receptors for ligand binding prediction using the molecular docking method. The 3D structure of diffractaic acid was constructed and structurally optimized using GaussView 6.0.16 and Gaussian 16 with the 6-31G* basis set [39]. Each target performed molecular docking using AutoDock Vina 1.1.2 for 100 runs [40,41,42] to predict the preferential binding site of diffractaic acid. Subsequently, the top pose from each run was chosen to compare with the binding energy of native inhibitor of each viral target to identify the promising target for diffractaic acid. The top binding pose was used to further elucidate the interaction of diffractaic acid/NS5 MTase using the Discover Studio Visualizer [43].

### 4.11. Dengue Methyltransferase (MTase) Assay

The MTase assay for compound inhibition testing was adapted and optimized as previously described [44]. Each reaction was performed in 50 µL of reaction using the MTase from DENV-2. Then, 10 µL of the enzyme complex (100 µg/mL MTase, 1.25 µM capped RNA substrate, and 10× MTase-Glo reagent (Promega, Madison, WI, USA) in 4× reaction buffer (80 mM Tris–HCl pH 8.0, 200 mM NaCl, 4 mM EDTA, 12 mM MgCl_2_, 0.4 mg/mL BSA, and 4 mM DTT)) was mixed with 5 µL of various concentrations of diffractaic acid and incubated at room temperature for 10 min. Subsequently, 10 µL of 2.5 µM S-adenosylmethionine (SAM) was added and incubated at 37 °C for 10 min. After incubating, 25 µL of MTase-Glo detection solution was added and incubated at room temperature for 60 min. The luminescence was measured using a VICTORTM X3 2030 Multilabel Reader (Perkin Elmer, Waltham, MA, USA). The percentage enzyme activity was calculated from two independent experiments using PRISM version 9 (GraphPad Software, La Jolla, CA, USA).

## 5. Conclusions

The antiviral activities of a lichen metabolite, diffractaic acid, against DENV1–4 and ZIKV, were reported for the first time. The cell-based studies and molecular modeling suggested the possible targets of the virus and the host components involved in viral replication, likely the viral methyltransferase and other targets. Further studies are required to identify the target molecules.

## Figures and Tables

**Figure 1 molecules-28-00974-f001:**
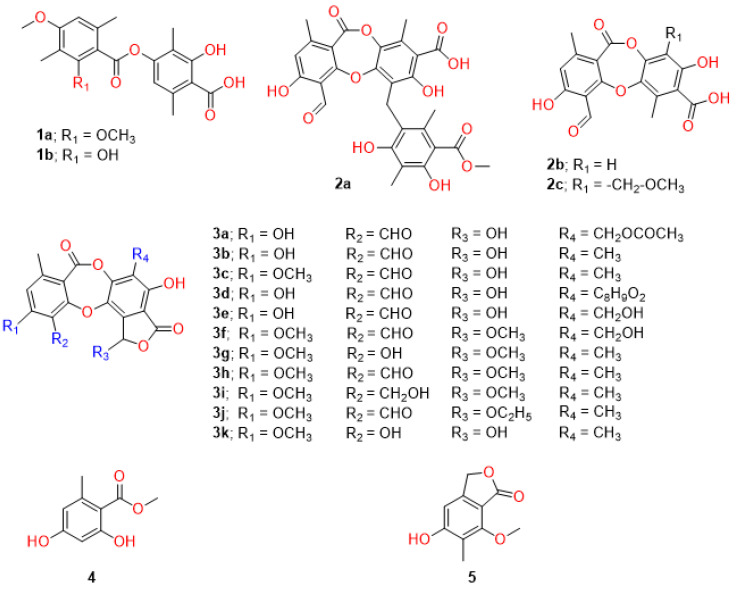
Chemical structures of depsides, depsidones, and their substructures.

**Figure 2 molecules-28-00974-f002:**
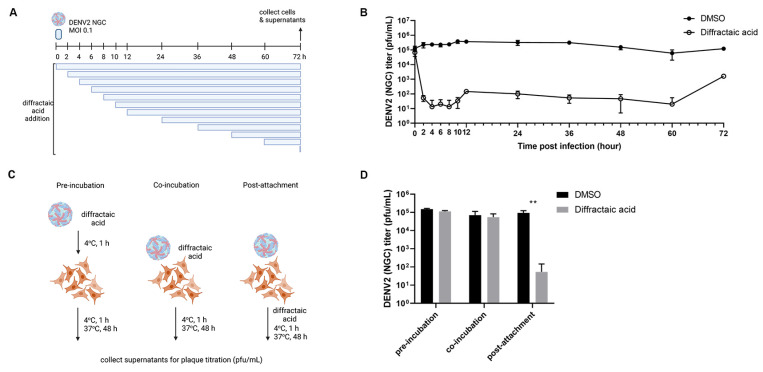
Time course study of diffractaic acid. (**A**) Schematic and (**B**) plaque titer results the time-of-addition assay. (**C**) Schematic and (**D**) plaque titer results of the anti-attachment assay. The DMSO-treated samples were used as no inhibition control; ** *p* < 0.01.

**Figure 3 molecules-28-00974-f003:**
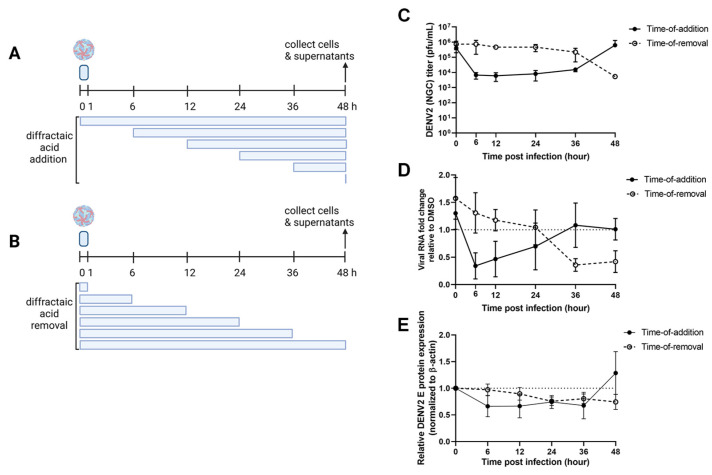
Effect of diffractaic acid on DENV2 virion, RNA, and protein expression. (**A**,**B**) Schematic of the perion of compound incubation (blue rectangles) in DENV2-infected Vero cells (MOI of 1). (**C**) Supernatants of the compound-treated samples were collected for DENV2 plaque titration. (**D**) Viral RNA was quantified by RT-qPCR and calculated as the fold change expression relative to DMSO-treated control. (**E**) Cell lysates were subjected to SDS-PAGE and Western blot to quantify DENV2 E protein expression. The anti-β-actin antibody was used as the loading control. The relative band intensities of DENV2 E protein from three independent experiments were calculated. The dotted line at 1.0 (y-axis) in (**D**,**E**) represents the equal expression of compound treatment compared to DMSO control.

**Figure 4 molecules-28-00974-f004:**
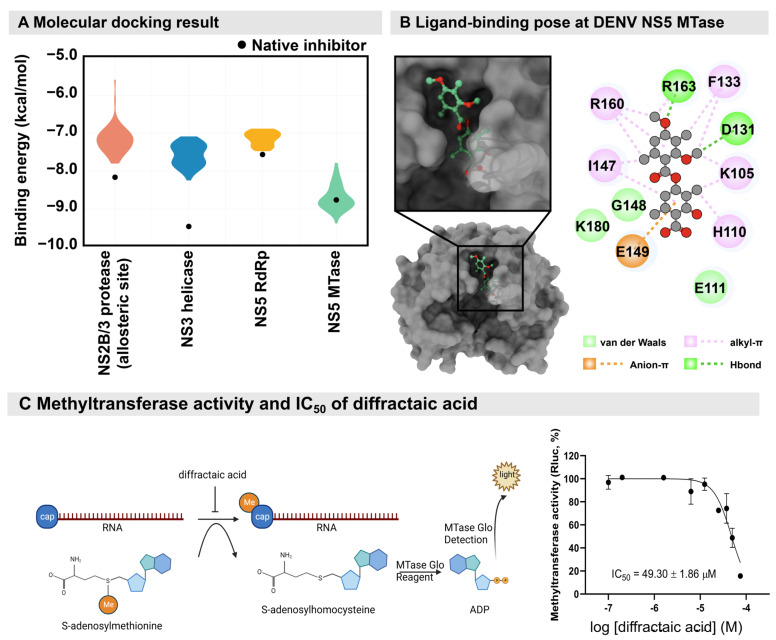
(**A**) Molecular docking of diffractaic acid toward DENV viral proteins; NS2/3 protease at allosteric site (salmon), NS3 helicase (blue), NS5 RdRp (yellow), and NS5 MTase (green), compared to native inhibitors of each target (black dot). (**B**) The preferential binding conformation of diffractaic acid (green) in complex with NS5 MTase (gray) presented as the ligand conformation and 2D interaction. (**C**) Schematics of the methyltransferase assay and an inhibitory concentration (IC_50_) of diffractaic acid.

**Table 1 molecules-28-00974-t001:** Antiviral activities and cytotoxicities of depsides, depsidones, and their substructures.

Code	IUPAC Name	Common Name	Biological Source	Viral Inhibition		Cell Viability		SI
1		Depsides		DENV2 NGC (%) ^1^	EC_50_ (µM) ^2^	Vero (%) ^1^	CC_50_ (µM) ^2^	
**1a**	4-((2,4-dimethoxy-3,6-dimethylbenzoyl)oxy)-2-hydroxy-3,6-dimethylbenzoic acid	Diffractaic acid	*Usnea aciculifera*	99.98 ± 0.04	DENV2 (NGC) 2.43 ± 0.19 DENV2 (16681) 3.89 ± 0.07 DENV1 (16007) 4.57 ± 0.26 DENV3 (16562) 4.90 ± 0.49 DENV4 (1036) 4.43 ± 0.58 ZIKV (SV0127/14) 2.83 ± 0.50 CHIKV (ECSA) 6.21 ± 0.69 EV-A71 (BRCR) 19.93 ± 1.94	114.72 ± 5.88	Vero 50.13 ± 7.45 Huh-7 >100 HepG2 >100 HEK-293 70.44 ± 0.33 THP-1 49.60 ± 1.06 RD 64.34 ± 5.04	DENV2 (NGC) 20.59 DENV2 (16681) >25 DENV1 10.97 DENV3 10.23 DENV4 11.33 ZIKV 17.71 CHIKV 8.07 EV-A71 3.32
**1b**	2-hydroxy-4-((2-hydroxy-4-methoxy-3,6-dimethylbenzoyl)oxy)-3,6-dimethylbenzoic acid	Barbatic acid	*Usnea aciculifera*	99.99 ± 0.02	DENV2 (NGC) 0.91 ± 0.15	113.31 ± 7.29	Vero 12.10 ± 0.38	DENV2 (NGC) 13.33
**2, 3**		**Depsidones**						
**2a**	6-(2,4-dihydroxy-5-(methoxycarbonyl)-3,6-dimethylben-zyl)-4-formyl-3,7-dihydroxy-1,9-dimethyl-11-oxo-11H-dibenzo[b,e][1,4]dioxepine-8-carboxylic acid	Parmosidone C	*Parmotrema tsavoense*	96.48 ± 2.90	DENV2 (NGC) 17.42 ± 3.21	123.58 ± 17.10	Vero 155.83 ± 7.77	DENV2 (NGC) 8.95
**2b**	4-formyl-3,8-dihydroxy-1,6-dimethyl-11-oxo-11H-dibenzo[b,e][1,4]dioxepine-7-carboxylic acid	Subvirensic acid	*Usnea baileyi*	Not inhibited		114.15 ± 15.82		
**2c**	4-formyl-3,8-dihydroxy-9-(methoxymethyl)-1,6-dimethyl-11-oxo-11H-dibenzo[b,e][1,4]dioxepine-7-carboxylic acid	9′-O-methylprotocetraric acid	*Usnea baileyi*	Not inhibited		102.53 ± 3.32		
**3a**	(11-formyl-1,4,10-trihydroxy-8-methyl-3,7-dioxo-1,3-dihydro-7H-benzo[6,7][1,4]dioxepino[2,3-e]isobenzofuran-5-yl)methyl acetate	Galbinic Acid	*Parmotrema dilatatum*	80.50 ± 4.85		112.03 ± 3.55		
**3b**	1,4,10-trihydroxy-5,8-dimethyl-3,7-dioxo-1,3-dihy-dro-7H-benzo[6,7][1,4]dioxepino[2,3-e]isobenzofuran-11-carbaldehyde	Norstictic acid	*Usnea aciculifera*	78.00 ± 7.55		111.18 ± 5.63		
**3c**	1,4-dihydroxy-10-methoxy-5,8-dimethyl-3,7-dioxo-1,3-dihydro-7H-benzo[6,7][1,4]dioxepino[2,3-e]isobenzofuran-11-carbaldehyde	Stictic acid	*Usnea aciculifera*	Not inhibited		108.02 ± 4.34		
**3d**	1,4,10-trihydroxy-8-methyl-3,7-dioxo-5-(2,4,6-trihydroxyben-zyl)-1,3-dihydro-7H-benzo[6,7][1,4]dioxepino[2,3-e]isobenzofuran-11-carbaldehyde	Parmosidone F	*Parmotrema dilatatum*	Not inhibited		120.97 ± 12.71		
**3e**	1,4,10-trihydroxy-5-(hydroxymethyl)-8-methyl-3,7-dioxo-1,3-dihydro-7H-benzo[6,7][1,4]dioxepino[2,3-e]isobenzofuran-11-carbaldehyde	Salazinic acid	*Parmotrema dilatatum*	Not inhibited		120.10 ± 6.87		
**3f**	4-hydroxy-5-(hydroxymethyl)-1,10-dimethoxy-8-methyl-3,7-dioxo-1,3-dihydro-7H-benzo[6,7][1,4]dioxepino[2,3-e]isobenzofuran-11-carbaldehyde	8′-O-methylconstictic acid	*Usnea baileyi*	Not inhibited		112.63 ± 17.52		
**3g**	4,11-dihydroxy-1,10-dimethoxy-5,8-dimethyl-7H-benzo[6,7][1,4]dioxepino[2,3-e]isobenzofuran-3,7(1H)-dione	8′-O-methylmenegazziaic acid	*Usnea baileyi*	Not inhibited		122.40 ± 4.58		
**3h**	4-hydroxy-1,10-dimethoxy-5,8-dimethyl-3,7-dioxo-1,3-dihydro-7H-benzo[6,7][1,4]dioxepino[2,3-e]isobenzofuran-11-carbaldehyde	Methylstictic acid	*Usnea baileyi*	Not inhibited		118.00 ± 4.69		
**3i**	4-hydroxy-11-(hydroxymethyl)-1,10-dimethoxy-5,8-dimethyl-7H-benzo[6,7][1,4]dioxepino[2,3-e]isobenzofuran-3,7(1H)-dion	8′-O-methylcryptostictic acid	*Usnea baileyi*	Not inhibited		108.59 ± 5.75		
**3j**	1-ethoxy-4-hydroxy-10-methoxy-5,8-dimethyl-3,7-dioxo-1,3-dihydro-7H-benzo[6,7][1,4]dioxepino[2,3-e]isobenzofuran-11-carbaldehyde	8′-O-ethylstictic acid	*Usnea baileyi*	Not inhibited		97.88 ± 5.73		
**3k**	1,4,11-trihydroxy-10-methoxy-5,8-dimethyl-7H-benzo[6,7][1,4]dioxepino[2,3-e]isobenzofuran-3,7(1H)-dione	Menegazziaic acid	*Usnea baileyi*	Not inhibited		92.00 ± 11.41		
**4**	methyl 2,4-dihydroxy-6-methylbenzoate	Methyl orselinate	*Usnea aciculifera*	Not inhibited		106.62 ± 8.05		
**5**	5-hydroxy-7-methoxy-6-methylisobenzofuran-1(3H)-one	7-hydroxy-5-methoxy-6-methylphthalide	*Usnea aciculifera*	Not inhibited		119.68 ± 11.66		

^1^ Means and standard error of the mean (SEM) of one experiment with triplicate results. ^2^ Means and standard error of the mean (SEM) of three independent experiments.

## Data Availability

Not applicable.

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
