# Peer review of "Lichen-Derived Diffractaic Acid Inhibited Dengue Virus Replication in a Cell-Based System"

_molecules, 2023, doi:10.3390/molecules28030974_

Round 1
Reviewer 1 Report (Previous Reviewer 1)
The present work highlights on the discovery of a lichen-derived depside (diffractaic acid) as a new dengue virus inhibitor. Generally, it seems that all all experimental methods used in this work are reasonable. However, several following key points should be noted and improved:
1-In bioassay screening using DENV-2 infected cells, the native inhibitor (sinefungin) should be used as positive control. But it has different chemical structure with diffractaic acid and is used to block the methylation of bases in DNA and RNA.
2-Although TOA and TOR assays were successfully conducted, the real target of diffractaic acid is not yet determined by molecular docking analysis. Therefore, further study on its methyltransferase activity of should be performed to confirm the mechanism of action.
3-Chemical isolation of compounds 1a, 1b, 3b, 4 and 5 were described in section 4.1. How about other substances? And their NMR and HRESIMS spectra together should be supplemented in supporting materials.
Author Response
The present work highlights on the discovery of a lichen-derived depside (diffractaic acid) as a new dengue virus inhibitor. Generally, it seems that all experimental methods used in this work are reasonable. However, several following key points should be noted and improved:
1-In bioassay screening using DENV-2 infected cells, the native inhibitor (sinefungin) should be used as positive control. But it has different chemical structure with diffractaic acid and is used to block the methylation of bases in DNA and RNA.
Thank you for the comments. In fact, one of the native inhibitors of dengue methyltransferase is sinefungin (Noble, CG, 2013; Lim SP, 2008; PDB 4R8S). The dengue methyltransferase enzyme methylates the guanine cap (N7 MTase) and the first nucleotide (2’-O MTase) of the viral RNA.
2-Although TOA and TOR assays were successfully conducted, the real target of diffractaic acid is not yet determined by molecular docking analysis. Therefore, further study on its methyltransferase activity of should be performed to confirm the mechanism of action.
Thank you for the suggestion. We performed the dengue methyltransferase assay and the IC50 was 49.30 ± 1.86 µM (Fig. 4c). The results were reported in (page 4, lines 196-199) and discussed in (page 5, lines 249-251).
3-Chemical isolation of compounds 1a, 1b, 3b, 4 and 5 were described in section 4.1. How about other substances? And their NMR and HRESIMS spectra together should be supplemented in supporting materials.
Details of the other chemical identities can be found in the previous publications (Refs 7-10).
7. Van Nguyen, K., et al., Chemical constituents of the lichen Usnea baileyi (Stirt.) Zahlbr. Tetrahedron Letters, 2018. 59(14): p. 1348-1351.
- Devi, A.P., et al., Salazinic Acid-Derived Depsidones and Diphenylethers with α-Glucosidase Inhibitory Activity from the Lichen Parmotrema dilatatum. Planta Medica, 2020.
- Duong, T.-H., W. Chavasiri, and J. Boustie, New meta-depsidones and diphenyl ethers from the lichen Parmotrema tsavoense (Krog & Swinscow) Krog & Swinscow, Parmeliaceae. Tetrahedron, 2015. 71(52): p. 9684-9691.
- Truong, T.L., et al., A new depside from Usnea aciculifera growing in Vietnam. Nat Prod Commun, 2014. 9(8): p. 1179-80.
Reviewer 2 Report (New Reviewer)
Pls see the attached file for comments.

Author Response
Thank you for reviewing our manuscript. According to the attached file, there are three comments as follows;
- Pl rephrase and add proper conclusion with novel findings.
Thank you for the suggestion. We added the proper conclusion at page 1, line 42-44. - Findings can be interesting to interpret by using an animal model.
Thank you for the suggestion. We discussed animal toxicity (LD50 962 mg/kg) in page 5, lines 215-216. - surface plasmon resonance (SPR) assay may be performed to confirm the binding region.
Thank you for the suggestion. We performed the enzymatic assay (methyltransferase activity) in the revised manuscript and discussed the results (page 4, lines 196-199).
Reviewer 3 Report (New Reviewer)
The manuscript “Lichen-derived diffractaic acid inhibited dengue virus replication in a cell-based system” (Loeanurit et al.) focussed on the antiviral activity of a natural compound (namely, diffractaic acid). After an initial screen on the anti DENV2 activity of a set of 18 compounds, the antiviral profile of compound 1a was deeply analysed and docking simulations suggest a potential molecular target for this molecule (namely, DENV NS5 MTase).
The following observations have been raised during the review:
The authors are encouraged to use institutional email addresses. Furthermore, in the affiliation list, the author should define their initials to be used in the “Author Contributions” section.
Line 48: reference is missing. Please indicate in the text the year for the reported statistics
Line 50: replace “was” with “is”
Line 69: replace “[7] [8] [9] [10]” with “[7-10]”. Please check reference formatting through the text.
Line 80: NGC acronym is not defined
Line 91: please delete the sentence “The higher selectivity index indicated the potentially broader therapeutic range in the animal study”. SI is not predictive of any in vivo property.
Figure 1. For sake clarity, please include the substituents of compounds 3 in a table. Moreover, in compounds 2, R1 substituent can be indicated in molecular structure as it is the same for all three compounds. The chemical names of compounds 4 and 5 appear to be redundant
Table 1: please reformat the table according to the journal indications and improve its clarity. Its reading is very difficult.
Line 112: please provide a rationale behind the cell line selection.
Lines 267-274: the reported consideration on COX-2 inhibition are not supported by any experimental evidence. Please delete
Line 422: The “Supplementary Materials” section is missing.
Please reformat the Reference list according to the journal indications
Author Response
The manuscript “Lichen-derived diffractaic acid inhibited dengue virus replication in a cell-based system” (Loeanurit et al.) focussed on the antiviral activity of a natural compound (namely, diffractaic acid). After an initial screen on the anti DENV2 activity of a set of 18 compounds, the antiviral profile of compound 1a was deeply analysed and docking simulations suggest a potential molecular target for this molecule (namely, DENV NS5 MTase). The following observations have been raised during the review:
- The authors are encouraged to use institutional email addresses. Furthermore, in the affiliation list, the author should define their initials to be used in the “Author Contributions” section.
Thank you for the suggestion. Miss Naphat Loeaurit and Miss Vipanee Vibulakhaophan do not possess institutional email addresses. Moreover, we replace full names with initials in the “Author Contributions” section. - Line 48: reference is missing. Please indicate in the text the year for the reported statistics
Thank you for the suggestion. The year 2022 is added. - Line 50: replace “was” with “is”
Thank you for the suggestion. We already changed the word. - Line 69: replace “[7] [8] [9] [10]” with “[7-10]”. Please check reference formatting through the text.
Thank you for the suggestion. We already changed the word. - Line 80: NGC acronym is not defined
Thank you for the suggestion. We already added the new guinea C strain (NGC) to the text. - Line 91: please delete the sentence “The higher selectivity index indicated the potentially broader therapeutic range in the animal study”. SI is not predictive of any in vivo property.
Thank you for the suggestion. We already adjusted the sentence. - Figure 1. For sake clarity, please include the substituents of compounds 3 in a table. Moreover, in compounds 2, R1 substituent can be indicated in molecular structure as it is the same for all three compounds. The chemical names of compounds 4 and 5 appear to be redundant.
Thank you for the suggestion. In our previous submission, we tried incorporating the structures into the table before but it was harder to follow. We were then suggested to change the table format into this version. We are totally willing to accommodate the reviewers as much as possible. Therefore, - Table 1: please reformat the table according to the journal indications and improve its clarity. Its reading is very difficult.
Thank you for the suggestion. In our previous submission, we were suggested to change the table format into this version. We also double checked with the journal website. In case of additional suggestion, please indicate the specific clarity improvement so we can make the right adjustment. - Line 112: please provide a rationale behind the cell line selection.
Thank you for the suggestion. We added the rationale behind the cell line selection as all of them were susceptible to the flaviviruses. Moreover, HepG2, HEK-293, and THP-1 cells represented hepatocytes, renal epithelia, and monocytes that were generally cellular targets after the primary viremia (page 3, lines 114-115) - Lines 267-274: the reported consideration on COX-2 inhibition are not supported by any experimental evidence. Please delete
Thank you for the suggestion. We already removed the paragraph. - Line 422: The “Supplementary Materials” section is missing.
Thank you for the suggestion. We already added the Supplementary Materials in another file according to the format of the journal. - Please reformat the Reference list according to the journal indications
Thank you for the suggestion. We updated the references’ format to MDPI.
Round 2
Reviewer 1 Report (Previous Reviewer 1)
As described in text, the molecular target of the lichen-derived diffractaic acid remains to be elucidated. It is the key point that the real mechanism of action should be unambiguously confirmed.
Author Response
As described in text, the molecular target of the lichen-derived diffractaic acid remains to be elucidated. It is the key point that the real mechanism of action should be unambiguously confirmed.
It is possible that sometimes the actual molecular target has not been confirmed such as anti-bacterial azithromycin, several antidepressants and anticancer drugs. We have explored as much as possible to identify the potential targets including computational screening and enzymatic assay validation, generating a revertant mutant and identifying by next-generation sequencing.
This manuscript is a resubmission of an earlier submission. The following is a list of the peer review reports and author responses from that submission.
Round 1
Reviewer 1 Report
The present work highlights the isolation and anti-flaviviral evaluation of 18 depsides and depsidones from several lichens as well as the analysis of mechanism of action (MOA). Despite of anti-flaviviral effect, it seems that it has no novelty. In additon, several following points should be noted:
1-Introduction: what is the status that anti-flaviviral drugs and candidates are developed? It is closely related with the efficacy of these lichen-derived depsides and depsidones.
2-Results: how to characterize these 18 depsides and depsidones? What is the positive control in biological tests? Why not use An-flavi, an important web platform to predict inhibitors of flaviviruse using QSAR and peptodomimetic approaches.
3-Methods: where are NMR and MS spectra of isolated compounds? And their chemical structures should be provided.
Reviewer 2 Report
This paper may provide interesting results but is very difficult to follow and to understand because some English sentences are not clear. Starting from the title (which I do not like at all), many sentences such as “Line 39 - Dengue virus (DENV1-4) causes a global burden estimated at 390 million people being at risk” or “Line 46 - The secondary infection from different serotypes leads to a risk of severity” have no meaning. Therefore, the first revision require a deep improvement of language.
Further examples of not clarity are the following:
Line 29: “Cytotoxicity concentrations (CC50)……” which is the meaning of this concentration?
Line 66: …..“showed over 90 % inhibition”…… inhibition of what? Similarly on line 83….
In the present form, it is impossible for me to review this manuscript.
Reviewer 3 Report
This is an interesting work on aromatic compounds extracted from lichens with antiviral activities and could be accepted for publication after the following revisions.
-please improve English level of your manuscript
-Abstract: better explain what you meant with ' properties despite anti-flaviviruses '
-Abstract: why do you declare that TT-031 is the most effective compound? TT-032 has an EC50 apparently lower. Specify already in the abstract if you consider it the most effective based on selectivity index etc
-Introduction: provide a Figure or Scheme with the generic molecular representation of depsides and depsidones
-lines 54-55: 'were reported to had', check English in all sections. Also other mistakes like this
-line 89: As you mention at pag 9 TT-031 could act as an anticancer drug. Are there any phase1/2 clinical studies on its employment in this field? May we say that TT-031 could be soon usedl for anticancer therapy or it is still far from being transferred into clinics? Please briefly comment on this
-Figure 1, (h): can you adjust scale for the last panel (h)? Same for Figure 2 (j)
-line 109: report SD deviations for the range 2.43-4.90
-Line 119: add a reference for using DMSO as a no inhibition control. Is it a standard methodology? Or was used simply being the solvent in which the compound was used? Specify here.
-line 169: the docking shows a preference for site 2. please explain what the 3 percentages shown stand for.
-Fig 6: provide IUPAC names for the two molecules
-sec. 5: Give more details in your Conclusions
